# Over half of clinical practice guidelines use non-systematic methods to inform recommendations: A methods study

**Carole Lunny**[1]*, **Cynthia Ramasubbu**[1], **Lorri Puil**[1], **Tracy Liu**[1], **Savannah Gerrish**[1], **Douglas M. Salzwedel**[1], **Barbara Mintzes**[2], **James M. Wright**[1]

**1** Department of Anesthesiology, Pharmacology & Therapeutics, Faculty of Medicine, Cochrane Hypertension Review Group, Therapeutics Initiative, University of British Columbia, Vancouver, BC, Canada, **2** Charles Perkins Centre, and School of Pharmacy, The University of Sydney, Camperdown, NSW, Australia

* carole.lunny@ti.ubc.ca

## Abstract

### Introduction

Assessing the process used to synthesize the evidence in clinical practice guidelines enables users to determine the trustworthiness of the recommendations. Clinicians are increasingly dependent on guidelines to keep up with vast quantities of medical literature, and guidelines are followed to avoid malpractice suits. We aimed to assess whether systematic methods were used when synthesizing the evidence for guidelines; and to determine the type of review cited in support of recommendations.

### Methods

Guidelines published in 2017 and 2018 were retrieved from the TRIP and Epistemonikos databases. We randomly sorted and sequentially screened clinical guidelines on all topics to select the first 50 that met our inclusion criteria. Our primary outcomes were the number of guidelines using either a systematic or non-systematic process to gather, assess, and synthesise evidence; and the numbers of recommendations within guidelines based on different types of evidence synthesis (systematic or non-systematic reviews). If a review was cited, we looked for evidence that it was critically appraised, and recorded which quality assessment tool was used. Finally, we examined the relation between the use of the GRADE approach, systematic review process, and type of funder.

### Results

Of the 50 guidelines, 17 (34%) systematically synthesised the evidence to inform recommendations. These 17 guidelines clearly reported their objectives and eligibility criteria, conducted comprehensive search strategies, and assessed the quality of the studies. Of the 29/50 guidelines that included reviews, 6 (21%) assessed the risk of bias of the review. The quality of primary studies was reported in 30/50 (60%) guidelines.

**Data Availability Statement:** Our raw data files are available from the Open Science Framework (DOI 10.17605/OSF.IO/8RXNP). All other data are

contained in the manuscript and Supporting Information files.

**Funding:** The authors received no specific funding for this work.

**Competing interests:** The authors have declared that no competing interests exist.

## Conclusions

High quality, systematic review products provide the best available evidence to inform guideline recommendations. Using non-systematic methods compromises the validity and reliability of the evidence used to inform guideline recommendations, leading to potentially misleading and untrustworthy results.

## 1.0 Introduction

Clinical practice guidelines (guidelines) help healthcare practitioners navigate the complexities in patient care, and facilitate informed, shared clinician-patient decision-making. Standards for guideline development are published by many organisations [1–3], and although there are slight differences, consensus exists surrounding key aspects of guideline development particularly regarding the use of systematic reviews to inform their development. Despite these standards and guidance, many guidelines do not conduct a systematic, evidence-based approach to knowledge synthesis [4, 5].

Generally, the development of guideline recommendations follows a series of steps [6], starting with the convening of a working group, conflict of interest management, and specification of the clinical questions and relevant outcomes. Research questions help define literature searches, inform the planning and process of the evidence synthesis, and act as a guide for the development of recommendations. Evidence synthesis is an integral part of guideline development and typically involves the following steps: specification of the purpose, objectives and scope of the review; specification of eligibility criteria and literature search methods; data extraction; assessment of risk of bias of included studies, synthesis of findings, and assessment of the quality or certainty of the evidence for each outcome. After the synthesis steps are completed, the working group translates the evidence into recommendations. The higher the certainty of a body of evidence, the more likely a strong recommendation can be made. However, recommendations incorporate additional considerations such as the net balance of benefits and harms, values and preferences, resource use and acceptability [6].

Guideline developers may use some or all of these steps and various methods in the conduct of the evidence synthesis. One of several types of systematic evidence syntheses may be used. These include systematic reviews that synthesize the results of original primary studies (e.g. randomized trials, cohort studies), and systematic 'overviews of reviews'. The latter, also called umbrella reviews, reviews of reviews, or meta-reviews, synthesize the results of existing systematic reviews. Guideline developers may also conduct a non-systematic literature review (i.e. no systematic methods used), or non-systematic overviews of reviews. Moreover, guideline developers may search for, and include, a variety, and combination, of different study designs that have been collected in either a systematic or a non-systematic way.

Many international standards exist for guideline developers when conducting guidelines, including guidance from the Institute of Medicine (IOM) [7], Guidelines International Network (GIN) [1], the Scottish Intercollegiate Guidelines Network (SIGN) [8], the National Institute for Health and Care Excellence (NICE) [2], the Australian National Health and Medical Research Council (NHMRC) [3], and the World Health Organization (WHO) [9], to name a few. The GRADE Working Group provides one of the most rigorous approaches, a framework for assessing the certainty of a body of evidence in an evidence synthesis, then interpreting the evidence into recommendations, and judging the strength of the recommendations [10, 11].

Despite existing international standards, surveys of guidelines [12–15] indicate many are of moderate to low quality, as assessed by the Appraisal of Guidelines for Research & Evaluation Instrument (AGREE II) tool [18]. AGREE II is the most commonly applied methodological quality guideline tool worldwide [15]. The tool's third domain deals with the methodological and/or reporting quality of the evidence synthesis process in a guideline. The most recent systematic survey of 421 guidelines found that 33% scored low on this domain for "rigor of development" [15]. Although popular, the AGREE II tool is not designed to provide a comprehensive and thorough evaluation of the methodological rigor of evidence syntheses within guidelines. **Table 1** summarizes the international standards on how to conduct the evidence synthesis process for guidelines by the Institute of Medicine (IOM) [7], AGREE II and Guidelines International Network (GIN) [1].

If we are to improve clinical practice, the evidence underpinning guideline recommendations must be rigorously developed and evaluated [16, 17]. Non-systematic methodology to gather, appraise, and synthesise evidence may lead to biased results and over- or under-estimation of treatment effect estimates, which are especially harmful when used to support guideline recommendations [18]. For example, in 2016, the Canadian Association of Radiologists (CAR) issued a guideline calling for women with average breast cancer risk to begin screening mammography at age 40 [19], in contrast to US Preventive Services Task Force [20] or the American Academy of Family Physicians [21] recommendations published in the same year. The results from three randomized trials from 2010, 2014, and 2015 show that the risk of cancer is lower for women ages 40 to 44, and the risk of harm from screening (biopsies for false-positive findings, over diagnosis) is higher compared to women over 50 [22]. These three trials could have been included in the CAR guideline but were not. No methods for how the guideline was developed were found in the guideline itself, nor on the association's webpages. In this case, use of non-systematic methodology may have led to a potentially harmful guideline recommendation.

Assessing the process used to synthesise the evidence underpinning recommendations in guidelines enables knowledge users to determine the trustworthiness of the recommendations [23]. We therefore aimed to (a) assess whether systematic methods were used when synthesizing the evidence for guidelines; and (b) evaluate the type of systematic review (with or without pairwise and network meta-analysis) or overview cited in support of recommendations.

**Table 1. Guidance on the evidence synthesis process in guideline development from IOM, AGREE II, and GIN.**

| IOM | AGREE II | GIN |
|---|---|---|
| "Statements that include **recommendations**, intended to optimize patient care that are informed by **a systematic review of evidence and an assessment of the benefits and harms of alternative care options.**" | Domain 3: Rigor of Development<br>7. **Systematic methods were used** to search for evidence.<br>8. The criteria for selecting the evidence are clearly described.<br>9. The strengths and limitations of the body of evidence are clearly described.<br>11. The health benefits, side effects and risks have been considered in formulating the recommendations.<br>12. There is an **explicit link between the recommendations and the supporting evidence**.<br>13. The guideline has been externally reviewed by experts prior to its publication.<br>14. A procedure for updating the guideline is provided. | *6. Evidence Reviews*<br>Guideline developers **should use systematic evidence review methods** to identify and evaluate evidence related to the guideline topic. |

## 2.0 Methods

### 2.1 Design and protocol

Our methods protocol was previously published in BMJ Open [24], and in the Open Science Framework (https://osf.io/rju4f/). We adhered to guidance for systematic reviews for searching, study selection, data extraction, and critical appraisal [25]. We adapted the PRISMA checklist for reporting our methods study during publishing. Our raw data files have been uploaded to the repository Open Science Framework (https://osf.io/8rxnp/ with DOI 10.17605/OSF.IO/8RXNP). Our methods are described below and in greater detail in our protocol [24].

### 2.2 Search

We searched the Turning Research Into Practice (TRIP) and Epistemonikos databases for guidelines dated from January 1, 2017 to December 31, 2018. This time period was selected to limit the number of guidelines screened due to resource limitations. The "Broad syntheses" filter in Epistemonikos was selected for retrieval of guidelines (**S1 Appendix**). Epistemonikos scans the following databases for relevant content: Campbell Library; the JBI Database of Systematic Reviews and Implementation Reports; EPPI-Centre Evidence Library Cochrane Database of Systematic Reviews; PubMed; Embase; CINAHL (The Cumulative Index to Nursing and Allied Health Literature); PsycINFO; LILACS (Literatura Latinoamericana y del Caribe en Ciencias de la Salud); and DARE (Database of Abstracts of Reviews of Effects). TRIP recently migrated all content from AHRQ's Clinical Guidelines Clearinghouse (www.guidelines.gov), which lost funding on July 16, 2018 (Jon Brassey, personal communication, April 10, 2018). TRIP indexes guidelines from over 289 journals.

### 2.3. Study selection

References retrieved from TRIP and Epistemonikos were imported into a single EndNote file and de-duplicated. Subsequently, we randomly sorted the citations retrieved using Microsoft Excel's RAND function and, used a Microsoft Excel (2013) form to screen.

Screening to identify citations meeting our inclusion criteria was conducted independently by two authors, starting with the lowest random number, until 50 guidelines were included. We chose this sample size as it was large enough to include a variety of clinical conditions, and be feasible for two reviewers to extract and assess reporting in the time available to the research team. Authors pilot tested the screened form on ten studies to establish agreement in definitions of eligibility criteria. We discussed any discrepant decisions until consensus was reached, or with a senior author.

### 2.4 Eligibility criteria

In addition to a requirement for publication between January 1, 2017 and December 31, 2018, we defined clinical practice guidelines according to the following inclusion criteria:

- Focused on the management or treatment of any clinical condition. For example, included clinically focused guidelines may include recommendations for ways to prevent harms associated with therapy, lifestyle modifications, when to implement or adjust therapy, and choice of therapy including treatment combinations.

- Developed by a group or organisation (i.e. not authored by one person).

- Comprise at least two explicit recommendations for treatment or management of a condition.

- Describe their methodology in the main manuscript of the guideline or in auxiliary documents.

- Provide a reference list (i.e. a bibliography).

We included guidelines in any language, however, because we searched only TRIP and Epistemonikos, we retrieved only English language guidelines. We will only include the most recent update of a guideline if more than one report is found. We will include any supplementary files to the main guideline, including methods documents and published systematic reviews.

We will exclude guidelines without recommendations or that focus solely on screening or diagnosis. We will also exclude guidelines where:

- Full text is inaccessible.

- The design is for local use only (e.g. in a single health facility or single regional health service).

- The design is restricted to hospitalized patients or patients in long-term care facilities.

- Focuses on patterns of use of medications (e.g. guidance about adherence to medications) but not treatment choice.

## 2.5 Definitions of evidence synthesis types

We classified approaches to evidence synthesis according to the following definitions.

*Literature reviews or non-systematic reviews* are summaries of the literature on a particular topic that are not developed systematically.

**2.5.1 Systematic review.** A review of evidence is considered systematic if it reports, at a minimum:

- Clearly formulated research question using PICOS (participants, interventions, comparisons, outcomes, and study design);

- Detailed inclusion and exclusion criteria for all included study types;

- Search algorithm for at least one database (i.e. reported search terms and a full search in an appendix);

- Searched two or more databases and described the search in the main body of the manuscript (i.e. not only in the abstract); and

- Process for selecting studies (e.g. independent screening, number of authors).

Systematic or non-systematic reviews may contain one or more *pairwise meta-analyses* or *network meta-analyses*. A pairwise meta-analysis compares the effect estimates of two interventions or one intervention and placebo from head-to-head trials (or observational studies). *A network meta-analysis* uses both direct comparisons from head-to-head trials and indirect comparisons based on a common comparator to compare multiple interventions [26].

*An overview of reviews* identifies, includes, and synthesises the results and conclusions of secondary analyses (i.e. reviews, systematic reviews, guidelines, or health technology assessments) and may or may not have used systematic methods as outlined above [27–29].

## 2.6 Outcomes

The study's primary outcome consisted of the numbers and proportions of recommendations within guidelines that were based on the following types of evidence syntheses:

1) Systematic reviews without meta-analysis

2) Systematic reviews with pairwise meta-analysis

3) Systematic reviews with network meta-analyses

4) Overviews of systematic reviews

We also evaluated the number of guidelines using either a systematic or non-systematic process to gather, assess, and synthesise evidence (**Fig 1**).

The secondary outcomes, calculated as numbers or proportions, are:

5) Guidelines that cited a Cochrane review or overview

6) Guidelines that report using GRADE methodology

7) Guidelines that report using other systems evaluating the strength of the recommendation and type of tool used (e.g. American Heart Association [30])

8) Guidelines that report using a level of evidence system and type of system used

9) Currency of the guideline (calculated by the time from last search to full publication)

10) Guidelines that report conflicts of interest disclosures by authors

## 2.7 Data extraction

We first examined the guidelines to determine whether reviews or overviews were cited in the guideline's recommendations, and then evaluated the treatment or management recommendations citing each review type. Review types were literature reviews, systematic reviews with pairwise meta-analysis, systematic reviews with network meta-analysis, and overviews of reviews.

We developed a data extraction form in Microsoft Excel (2013). We piloted the form on 10 guidelines and then discussed discrepancies in extracted data to come to consensus and to standardise the coding. Two authors extracted data independently and discrepancies were discussed until resolved. A senior author arbitrated conflicts. After all data was compared and reconciled, a senior author checked that the data was consistently coded across similar or related items.

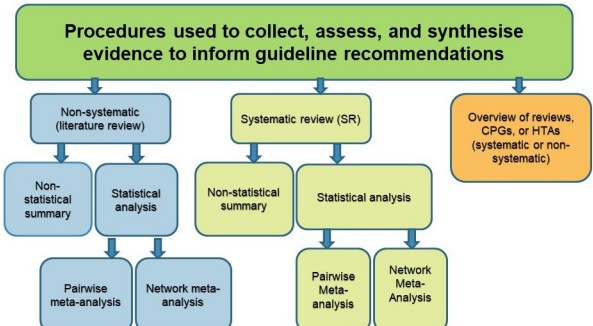

**Fig 1. Procedures used to collect, assess, and synthesise evidence to inform guideline recommendations (i.e. systematic, non-systematic).** Clinical practice guidelines can use a non-systematic or systematic process to collect, assess, and synthesise evidence to inform guideline recommendations. Guideline developers can conduct a (i) literature review (using non-systematic methods), (ii) systematic review (using systematic methods with inclusion of all eligible study types [e.g. primary studies, systematic reviews, overviews]), or (iii) an overview of systematic reviews (using either systematic or non-systematic methods with inclusion and synthesis of systematic reviews). Using these three evidence synthesis products, developers of guidelines can include only primary studies, both primary studies and systematic reviews, only systematic reviews, and/or both systematic reviews, clinical practice guidelines, health technology assessment (HTAs), or overviews of systematic reviews. This figure was adapted from Lunny et al. [24].

Guideline level data extracted included: our primary and secondary outcomes, name of the guideline, year of publication, country, the organisations or commissioning agency (publisher), type of publisher (government, medical society, university, other [specify]), aim, journal (if applicable), open source/paywall, the date of the last search, funding, declaration of conflicts of interest, stakeholder affiliation with/honoraria from pharmaceutical companies, target population (general population, or specific subpopulations such as those identified by age [e.g. children and adolescents, adults of any age, older adults], sex/gender, or co-morbidities), and scope (pharmacological or non-pharmacological treatment [e.g. surgical, medical device]).

We also evaluated whether critical appraisal of the review or overview was conducted, and recorded which tool was used (e.g. Assessing the Methodological Quality of Systematic Reviews [AMSTAR] 1 [31] or 2 [32], Risk of Bias Assessment Tool for Systematic Reviews [ROBIS] [33]).

## 2.8 Gaps in review-level evidence supporting a recommendation

If a guideline did not cite a Cochrane systematic review, we assumed the developers might have missed an important evidence synthesis. We therefore examined the Cochrane Database of Systematic Reviews using the terms and dates used in the search strategies of the guideline.

## 2.9 Evidence synthesis process in guidelines

To determine if a systematic process was used to gather, assess and synthesise evidence to inform recommendations, we used the following four criteria:

1) Clearly defined research questions or objectives reported in terms of PICOS (Populations, Interventions, Comparisons, Outcomes, and Study design) elements.

2) Clearly reported eligibility criteria for all included study designs.

3) Conducted a systematic search (i.e. two or more databases searched, keywords reported and a full search strategy reported in an appendix).

4) Reported a process for selecting/screening studies (e.g. independent process, number of authors).

We considered these criteria to be the minimum that can be used by a guideline to reduce bias and limitations when gathering evidence to inform recommendations. We also assessed whether the guideline working group reported the following methods:

• Assessment of the quality/risk of bias of the review or overview supporting/refuting the recommendation.

• Assessment of primary studies for quality/risk of bias.

These criteria were adapted from the ROBIS tool, which comprehensively assesses the risk of bias of a systematic review [33]. The tool includes items relating to risk of bias and classifies them according to study eligibility criteria; identification and selection of studies; data collection and study appraisal; and synthesis and findings.

## 2.10 Open access

All data management and study processes were conducted and recorded in the Open Science Framework.

## 2.11 Data analysis

The number and frequencies of citations of reviews and overviews in guidelines and their characteristics were calculated. We described and tabulated all primary and secondary outcomes. Additional information was put into appendices. We calculated the difference between the initial literature search date and publication date using the month and day function in Excel 2013 to estimate the time taken to conduct each guideline.

We performed a chi-square test of independence to examine the relation between using the GRADE approach and whether the guideline used a systematic process. Dependent categorical variables were type of organization (medical association, pharmaceutical, government, no funding, not reported), scope (narrow, broad), and continent (Europe, North America, Intercontinental). We also performed a chi-square test of independence to examine the relationship between GRADE use and type of funder, and guideline having conducted a systematic process and type of funder. We planned to explore whether the characteristics of guidelines differed in terms of pharmacological vs. non-pharmacological scope. However, there were too few guidelines with these characteristics to permit reliable comparisons ($\leq$10 in each group). We formally tested the associations using a chi-square test for one independent variable with 2 levels with categorical dependent variables in R.

## 3.0 Results

### 3.1 Search results

From 713 records retrieved from the TRIP and Epistemonikos databases, 691 remained after duplicate removal (**S2 Appendix** flowchart). The 691 records were then randomly sorted and screened sequentially. A total of 419 records were screened at full text to obtain our target of 50 eligible guidelines (see the list of included studies in **S3 Appendix**). Of the 369 excluded records, 47 guidelines were excluded as they did not have a methods section, and 16 did not include a reference section.

### 3.2 Characteristics of guidelines

The majority of the randomly selected guidelines were from Canada or the United States (31/50 [62%]) and were published in 2017 (40/50 [80%]; **Table 2**). Guidelines conducted in Europe constituted 32% (16/50). The most frequent medical condition addressed was malignant neoplasms (11/50 [22%]); however, the guidelines covered a broad range of clinical topics. Half of the guidelines (25/50 [50%]) had both a pharmacological and non-pharmacological scope. The average time from search to full publication was 24 months (range 2–204 months).

The three associations most frequently commissioning or conducting guidelines were the European Society for Medical Oncology (5/50 [10%]), the American Society of Clinical Oncology (4/50 [8%]), and the American Urological Association (3/50 [6%]). The majority of guidelines were funded by a medical society (18/50 [36%]), the pharmaceutical industry (9/50 [18%]), or funding was not reported (9/50 [18%]). A smaller number of guidelines were funded by government (8/50 [16%]), or did not receive any funding (6/50 [12%]).

The majority of guidelines were published in peer reviewed journals (42/50 [96%]), and were open access (45/50 [90%]). In 48 guidelines (96%), guideline authors declared their conflicts of interest, and in 33 (66%), authors declared affiliations with pharmaceutical companies (66%).

**Table 2. Characteristics of clinical practice guidelines (n = 50).**

| Characteristics | clinical practice guidelines n (%) |
|---|---|
| Year of publication | |
| 2017 | 40/50 (80) |
| 2018 | 10 (20) |
| Region* | |
| Canada or United States | 31 (62) |
| Europe | 16 (32) |
| All other regions | 3 (6) |
| Medical classification | |
| Malignant neoplasms | 11 (22) |
| Diseases of the cardiovascular system | 5 (10) |
| Diseases of the genitourinary system | 4 (8) |
| Endocrine, nutritional and metabolic diseases | 4 (8) |
| Diseases of the gastrointestinal system | 3 (6) |
| Mental and behavioural disorders | 3 (6) |
| Diseases of the skin and subcutaneous tissue | 2 (4) |
| Factors influencing health status and contact with health services | 2(4) |
| HIV/AIDS | 2 (4) |
| Symptoms, signs and abnormal clinical and laboratory findings | 2 (4) |
| Other classifications** | 23 (46) |
| Scope | |
| Both pharmacological and non-pharmacological | 25 (50) |
| Pharmacological only | 13 (26) |
| Non-pharmacological only (e.g. surgery, medical device) | 7 (14) |
| Funding of the association writing the guideline | |
| Medical society | 18 (36) |
| Pharmaceutical industry | 9 (18) |
| Government | 8 (16) |
| No funding received | 6 (12) |
| Not reported | 9 (18) |
| Publishing | |
| Published in a peer reviewed journal | 42 (84) |
| Open source publishing (open access) | 45 (90) |
| Conflict of interest | |
| Declaration of conflicts of interest by developers | 48 (96) |
| $\geq$1 stakeholder affiliated with pharmaceutical companies*** | 33 (66) |
| Average time from search to full publication (months [range]) | 24 (2–204) |

*Numbers do not total 50 as multiple categories can apply to one guideline.

**Ten additional topics were identified: complications of surgical and medical care (n = 1); congenital malformations (n = 1); diseases of oral cavity, salivary glands and jaws (n = 1); diseases of the blood (n = 1); diseases of the ear (n = 1); diseases of the eye (n = 1); diseases of the nervous system (n = 1); injuries to the abdomen, lower back, lumbar spine and pelvis (n = 1); osteopathies and chondropathies (n = 1); pregnancy and childbirth (n = 1).

*** Includes both employees of the pharmaceutical companies and consultant fees.

## 3.3 Approach to evidence synthesis in guidelines

According to our definition of a systematic process, as outlined in the methods, 17/50 (34%) of guidelines were systematic in their approach to evidence synthesis, and two thirds (33/50

[66%]) of guidelines were non-systematic (**Fig 2**). Of the 3/50 (6%) guidelines that used an overview method with the synthesis of systematic reviews, guidelines, or health technology assessments (HTAs), only one guideline was systematic in its approach to evidence synthesis, and two were non-systematic (**Table 3**).

Of the 17 guidelines using a systematic approach to evidence synthesis, 65% (11/17) used a qualitative synthesis approach, 18% used a pairwise meta-analysis (3/17), 12% (2/17) used a network meta-analysis approach, and one overview of systematic reviews used a qualitative synthesis approach (1/17) (**Table 3**). Of the three guidelines that conducted overviews, two (2/50 [4%]) did an overview of systematic reviews, and one (1/50 [2%]) did an overview of guidelines.

### 3.4 Specific methods used for evidence synthesis in guidelines

More than half of the guidelines (30/50 [60%]) used an explicit statement to develop guideline questions and/or objectives and structured these using the PICOS format (**Table 4**). Eligibility criteria were specified clearly in over half of the guidelines (29/50 [58%]), and a systematic search reporting keywords used was found in half of the guidelines (25/50 [50%]). A total of 31 guidelines searched two or more databases (31/50 [62%]), and a slightly lower number reported the process for selecting studies (27/50 [54%]).

Approximately one third of guidelines (17/50 [34%]) used the GRADE approach, and two thirds (33/50 [66%]) used another system to assess the strength of the evidence, such as the Guidelines Into Decision Support methodology [34] (**Table 4**).

Over half of guidelines (30/50 [60%]) reported assessing the quality of primary studies or reported using the GRADE approach (and therefore must have assessed the risk of bias of the primary studies although the assessments were not provided) (**Table 4**). Of these 30 guidelines, eight guidelines (8/30 [27%]) reported using the Cochrane risk of bias tool to assess the quality of randomised trials, 11 guidelines reported using another tool (11/30 [37%]) such as the Drug Effectiveness Review Project instrument [35], QUADAS 2 tool [36], or the Newcastle Ottawa

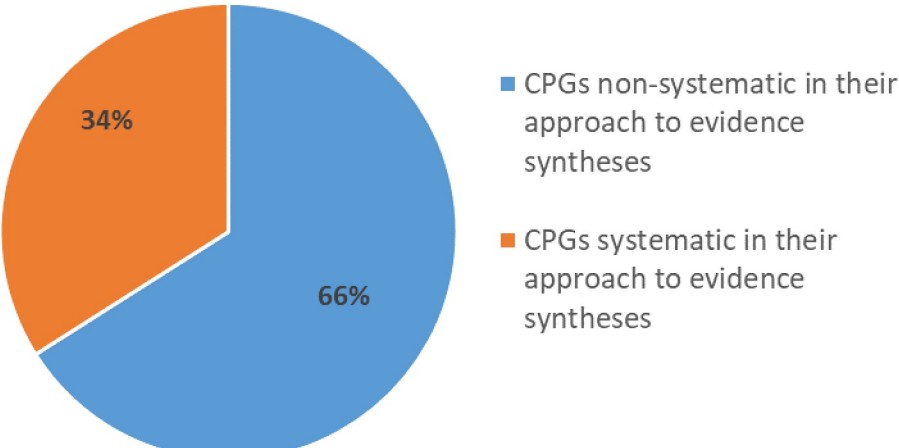

**Fig 2. Systematic or non-systematic process used by guidelines in their approach to evidence synthesis (n = 50).**
To determine if a systematic process was used to gather, assess and synthesise evidence to inform recommendations, we used the following four criteria: (1) explicit statement of the questions or objectives reported in terms of PICOS (Populations, Interventions, Comparisons, Outcomes, and Study design) elements; (2) eligibility criteria reported for all included study designs; (3) a systematic search conducted (i.e. two or more databases searched); and (4) process reported for selecting/screening studies (e.g. number of authors, independent process). We considered these criteria to be the minimum that can be used by a guideline to reduce bias and limitations when gathering evidence to inform recommendations.

**Table 3. Clinical practice guidelines using a systematic or non-systematic approach to evidence syntheses (n = 50).**

| Type of evidence synthesis | Guidelines n (%*) |
|---|---|
| **Reviews:** | **47 (94%)** |
| **Non-systematic reviews** | **31 (62%)** |
| • with qualitative synthesis | 22 (44%) |
| • with pairwise meta-analysis | 7 (14%) |
| • with network meta-analyses | 2 (4%) |
| **Systematic reviews** | **16 (32%)** |
| • with qualitative synthesis | 11 (22%) |
| • with pairwise meta-analysis | 3 (6%) |
| • with network meta-analyses | 2 (4%) |
| **Overviews:** | **3 (6%)** |
| **Non-systematic overviews** | **2 (4%)** |
| • of reviews with non-statistical summary | 1 (2%) |
| • of guidelines with non-statistical summary | 1 (2%) |
| **Systematic overview of reviews** | **1 (2%)** |
| • with non-statistical summary | 1 (2%) |

*Percentages do not sum to 100 as some guidelines used more than one approach.

scale [37], and 11 did not report the specific tool. Of the 29 guidelines that had defined eligibility criteria for inclusion of reviews, six (6/29 [21%]) assessed the risk of bias or quality using an appropriate tool like ROBIS [33] or AMSTAR 2 [32].

## 3.5 Assessment of whether reviews or overviews were used to inform recommendations

Of the 50 guidelines, 44/50 (88%) cited reviews to inform recommendations. There was a total of 128 recommendations citing 249 reviews of any type (**Table 5**). Of the cited

**Table 4. Specific methods used for evidence synthesis in guidelines.**

| Method | Guidelines n/N (%) |
|---|---|
| Formulation of the guideline question(s) or objective(s) in terms of PICOS elements | 30/50 (60) |
| Inclusion and exclusion criteria of studies reported | 29/50 (58) |
| Systematic search strategy | 25/50 (50) |
| Two or more databases searched | 31/50 (61) |
| Process reported for selecting/screening studies | 27/50 (54) |
| Assessment of methodological quality (risk of bias) of review/overview | 5/50 (10) |
| Assessment of methodological quality (risk of bias) of primary studies | 30/50 (60) |
| Cochrane risk of bias tool for randomized trials | 8/30 (27) |
| Other tool | 11/30 (37) |
| GRADE used but risk of bias methods not reported | 11/30 (37) |
| GRADE approach reported | 17/50 (34) |
| Other system for assessing the strength of recommendation reported | 33/50 (66) |
| Highest "level of evidence" rating | |
| High quality SR/MAs | 17/50 (34) |
| SR/MAs | 9/50 (18) |
| High quality RCTs | 13/50 (26) |
| RCTs | 4/50 (8) |

**Table 5. Type of review used in 128 recommendations.**

| Type of review used in the 128 recommendations | Reviews (n = 249) n (%) |
|---|---|
| **Non-systematic reviews:** | *59 (24%)* |
| • without meta-analysis | 30 (12%) |
| • with pairwise meta-analysis | 29 (12%) |
| **Systematic reviews:** | *190 (76%)* |
| • without meta-analysis | 23 (9%) |
| • with pairwise meta-analysis | 160 (64%) |
| • with network meta-analyses | 7 (3%) |

reviews, 160/249 (64%) were systematic reviews with pairwise meta-analysis, 7/249 (3%) were systematic reviews with network meta-analysis, and 23/249 were systematic reviews without meta-analysis (**Table 5**). Of the 190 systematic reviews to inform recommendations, 47/190 (25%) of these were Cochrane systematic reviews, representing 47/249 (19%) of all reviews.

Of the 45/50 (90%) guidelines that cite either a review or overview to inform recommendations, only 29/50 (58%) guideline developers specified in their eligibility criteria that reviews or overviews were included.

### 3.6 Gaps in evidence supporting a recommendation

Of the 50 guidelines, 16/50 (32%) cited a Cochrane systematic review or a Cochrane overview. Of the 34 remaining guidelines, 27/34 (79%) guidelines could have used and cited Cochrane reviews to inform the recommendations. The median number of Cochrane reviews that could have been cited based on the guidelines search strategy was one [range 0–18].

### 3.7 Potential associations between use of GRADE, systematic review process, and type of funder

A chi-square test of independence was performed to examine the association between use of the GRADE framework and having used a systematic process for evidence synthesis. No association was found between these variables, $X^2$ (1, N = 50) = 0.023, p = 0.9. Guidelines that used the GRADE framework were not more likely to have used a systematic process.

We also explored whether a relation existed between GRADE use and type of funder ($X^2$ [4, N = 50] = 9.05, p = 0.05), conflict of interest ($X^2$ [1, N = 50] = 0.18, p = 0.07), scope ($X^2$ [1, N = 50] = 1.4, p = 0.2), affiliation with the pharmaceutical industry ($X^2$ [1, N = 50] = 2.4, p = 0.11), and continent ($X^2$ [2, N = 50] = 6, p = 0.05). Upon further exploration, guidelines using GRADE were more likely to have been funded by government or the pharmaceutical industry, and conducted internationally (with an organisation like the WHO).

We also tested the relation between the guideline having conducted a systematic process and type of funder ($X^2$ [4, N = 50] = 3.60, p = .46), conflict of interest ($X^2$ [1, N = 50] = 0.9804, p = 0.322), scope ($X^2$ [1, N = 50] 0, p = 1.0), affiliation with the pharmaceutical industry ($X^2$ [1, N = 50] = 0.61, p = .43), and continent ($X^2$ [2, N = 50] = 7.55, p = .02). Guidelines that used a systematic process were more likely to have been conducted internationally (with organisation like the WHO). A table of these associations is found in **S4 Appendix**.

### 4.0 Discussion

In this sample, only a minority of guidelines systematically synthesised the evidence to inform recommendations, notably the guidelines by the WHO [38], the UK National Institute for

Health and Care Excellence (NICE) [39, 40] and the Thoracic Society of Australia and New Zealand [41]. These guidelines explicitly and clearly reported their objectives and eligibility criteria, conducted comprehensive search strategies, and assessed the methodological quality of the studies included in the review of the evidence. High quality, systematic review products produced by guideline working groups following established guidance provide the best available evidence to inform recommendations [42].

Two thirds, or 66%, of guidelines reported non-systematic methods to develop their recommendations. This percentage is likely an underestimation because we excluded some guidelines when selecting studies. A total of 47 guidelines (47/691 records [7%]) were excluded because they did not contain a methods section, and 16 were excluded (16/691 [2%]) because they did not cite any references. This is a small improvement from an assessment of guidelines done two decades ago [43], which stated that only 20% of guidelines specified search methods, and 25% did not cite any references.

Several possible explanations exist for why guideline developers may use non-systematic methods when reviewing the evidence to inform recommendations. First, guideline working groups may lack the required resources to undertake a full systematic process, which is time consuming and labour intensive. Second, guideline developers may be unaware of the importance of using systematic methods to minimise bias and error when synthesising evidence, and/or be unaware of the guidance available from evidence synthesis organisations, such as Cochrane [25], the GRADE Working Group [44], and other organisations [8, 33]. Some organisations and societies require that guideline developers adhere to established methodological standards (e.g. NICE [45]) and undergo mandatory training in these methodologies (e.g. Infectious Diseases Society of America [46]). However, guidance provided by other medical associations and societies on how to gather, appraise and synthesise evidence to inform recommendations can vary and may not adequately emphasize the steps needed to minimise biases [47].

Second, guideline developers' opinions may outweigh or ignore relevant evidence when formulating recommendations. Indeed, prior evaluations of clinical guidelines in a range of clinical specialty areas have found that many recommendations are based on expert opinion [48–51]. Expert opinion may appropriately be combined with empirical evidence in clinical practice, and in the absence of relevant research, expert opinion may be considered the best available evidence. The process used for making practice recommendations, including the role of expert opinion in decision making and reaching consensus, should be transparently reported. Caution is advised in situations when a non-systematic process for synthesising evidence to inform recommendations is used, as relevant literature may not be found, and research can be selected to confirm expert opinion. This potentially allows for self-serving biases, such as confirmation bias (selective gathering of, or ignoring, evidence), consensus bias (believing that one's opinions are relatively common and justified), and bias associated with conflicts of interest [52].

Our investigation confirms previous findings that investigators often fail to cite and use earlier research when preparing new research [53–56]. Many guideline developers, such as the WHO and the NHMRC, recommend the use of systematic reviews and overviews to underpin guideline recommendations [9]. The majority of guidelines in our sample used reviews to underpin recommendations, but only about a fifth cited Cochrane reviews. About 80% of guidelines could have cited a Cochrane review but did not. Citing Cochrane reviews is important as empirical evidence shows they are conducted more rigorously than non-Cochrane reviews [57–60]. Our findings are similar to other studies that identified 40–70% of guideline recommendations did not use or cite all the relevant Cochrane reviews [61–63]. Recommendations that are not based on review-level evidence may indicate problems with the guideline

methodology (e.g. search strategy [missing relevant systematic reviews]) or gaps in the evidence base (i.e. a lack of adequately designed relevant studies).

Overviews, which synthesize systematic reviews, guidelines, and health technology assessments, may be particularly useful when guideline developers are required to make decisions about which of a number of alternate treatments are the most effective and safe interventions to use [3]. Guidance has recently been developed to aid guideline developers in synthesizing the results of multiple systematic reviews and in reporting overviews [27–29]).

Evaluating how well a study has been conducted is essential to determine if the findings are trustworthy and relevant to patient care and outcomes. Two thirds of guideline developers in our sample of guidelines did not assess the risk of bias (quality) of included primary studies, and one fifth did not assess the methodological quality of included reviews. Risk of bias assessment is about identifying systematic flaws, bias or limitations in the design, conduct, or analysis of research. If a systematic review or an overview is at risk of bias and the guideline fails to assess this, the findings can be misleading. Several studies have shown that bias can obscure up to 60% of the real effect of a treatment [64, 65]. Evidence shows biased results from poorly designed and reported studies can mislead decision-making in healthcare at all levels [66–69].

Significant improvement is needed in the reporting of methods in guidelines. Efforts to improve reporting are underway with the publication of the RIGHT statement for reporting clinical practice guidelines [70]. Two of the 34 items in the RIGHT checklist ask about systematic review methods, namely:

- Whether the guideline is based on new systematic reviews done specifically for this guideline or whether existing systematic reviews were used (item 11a).

- If existing systematic reviews were used, reference these and describe how those reviews were identified and assessed (provide the search strategies and the selection criteria, and describe how the risk of bias was evaluated) and whether they were updated (item 11b).

The RIGHT statement, while providing a minimum standard of what should be reported in a clinical practice guideline, also goes further by suggesting methods that should have been conducted (i.e. doing a systematic review, or used existing systematic review). These reporting recommendation will help knowledge users identify what processes were followed by guideline developers to gather, assess and synthesise evidence.

Improved reporting will help users of guidelines assess the methodological quality of the evidence synthesis process used to inform recommendations. While the length of guidelines is often unwieldy, links in text to the full methodology should be provided. As with clinical trials and systematic reviews, all guidelines should conduct a priori protocols, plans, or registered reports [71]. Without these pre-specified methods and plans, knowledge users will not be able to assess selective reporting of outcomes, or selective handling of multiple measures or analyses. At a minimum, guideline developers should develop explicit research questions, define all outcomes within the domains of interest, and pre-specify plans for handling many different outcomes, measures, and analyses. Prospective registration of guidelines would promote transparency, help reduce potential for bias, and would serve to avoid unintended duplication of multiple guidelines on the same topic and in similar settings [72–74].

## 4.1 Implications for clinical care

Systematically developed evidence syntheses in guidelines provide the high quality evidence base that is needed to inform recommendations. Using non-systematic methods compromises the validity and reliability of the evidence used to inform guideline recommendations, leading to potentially misleading and untrustworthy results. Patients, healthcare providers and policy

makers need in turn the highest quality guidelines to help guide decisions about which treatments should be used in healthcare practice. Even weak or conditional guideline recommendations (e.g. in the context of sparse evidence) must be based on systematic methodology for the guideline to be trusted, and for appropriate therapeutic decisions to be made. The consequences of providing patient care and rolling out population health policies from results of guidelines that are based on non-systematic evidence syntheses is compromised patient care and safety [75].

## 4.2 Strengths and limitations

The strengths of our methods include a pre-specified study protocol, the adoption of systematic and transparent methods, specific and explicit eligibility criteria, broad search strategies, randomised selection of guidelines, and duplicate and independent processes for guideline selection and data extraction.

We randomly sampled guidelines from TRIP and Epistemonikos databases from 2017 and 2018. We believe, even though our search range is narrow and outdated by 3 years that the reporting and methods used in guidelines would not have changed substantially. Notably no changes have been made to the AGREE II tool which is used to assess guideline reporting, quality and processes [76]. However, as we only included guidelines published between 2017 and 2018, in English, and indexed in TRIP and Epistemonikos, our findings can neither be generalized to CPGs published in another time period, language or to guidelines indexed elsewhere.

To mitigate the subjectivity of classifying, coding characteristics and methods used in reporting guideline recommendations, all data extractors piloted the items on ten studies. The piloting results were discussed to refine the wording of the items, come to consensus about definitions, and calibrate the coding. We also used minimum criteria to assess the evidence synthesis process used by the guidelines. A full assessment of the biases in an evidence synthesis would involve using a methodological quality assessment tool for reviews like ROBIS [33] or AMSTAR-2 [32]. Reporting was inadequate across guidelines, thus limiting our assessment of the methodological quality of the evidence gathering, appraising, and synthesising process. A further limitation is that our study is focused on guidelines for the management or treatment of any clinical condition [24].

By searching for missed Cochrane evidence, we evaluated whether a guideline might be missing high quality evidence. Cochrane reviews are known for using robust methodology [57–60]. A final limitation is that we may have missed high quality reviews by not searching for 'non-Cochrane' reviews. However, Cochrane reviews can also be prone to biases and should not be considered high quality without assessment of the risks of bias.

## 4.3 Future research

Future studies looking into the use of reviews in screening or diagnostic recommendations would be useful to determine the quality of the evidence synthesis process in guidelines. Our assessment might also form a baseline of the completeness of reporting (prior to the release of the RIGHT standard for reporting clinical practice guidelines) against which any future assessments can be compared.

## 5.0 Conclusions

When evaluating a random sample of recent guidelines, we found that 66% of the guidelines did not use a systematic process to gather, appraise, and synthesise the evidence to inform recommendations. It is important for health care practitioners to appreciate this major limitation

of guidelines. Findings from our study can inform efforts to improve the processes used in guidelines to do an evidence synthesis. Significant improvement is needed in the conduct as well as the reporting of evidence synthesis methods in guidelines. Guideline developers should use the systematic methods endorsed by reputable evidence synthesis organisations.

This methods study is one of only a few studies to assess the use and citation of systematic reviews with or without pairwise meta-analysis, systematic reviews with network meta-analyses, and 'overviews of systematic reviews' in guidelines. The majority of guidelines in our sample used reviews to underpin recommendations, but only a fifth cited Cochrane reviews.

Systematic evidence syntheses in guidelines provide the high quality evidence base that is needed to inform recommendations. Using non-systematic methods compromises the validity and reliability of the evidence used to inform guideline recommendations, leading to potentially misleading and untrustworthy results. A systematic process should be followed to ensure the evidence synthesis is accurate, valid, of the highest methodological quality, and based on all of the eligible scientific information. Patients, healthcare providers and policy makers need in turn the highest quality guidelines to inform decisions about which treatments should be used in healthcare practice.

## Supporting information

**S1 Appendix. Search strategies.**
(DOCX)

**S2 Appendix. Flowchart of the study selection process.**
(DOCX)

**S3 Appendix. Included clinical practice guidelines.**
(DOCX)

**S4 Appendix. Table of associations.**
(DOCX)

## Author Contributions

**Conceptualization:** Carole Lunny.

**Data curation:** Carole Lunny, Cynthia Ramasubbu, Tracy Liu, Savannah Gerrish, Douglas M. Salzwedel.

**Formal analysis:** Carole Lunny.

**Investigation:** Carole Lunny.

**Methodology:** Carole Lunny.

**Project administration:** Carole Lunny.

**Supervision:** Carole Lunny, Lorri Puil, James M. Wright.

**Visualization:** Carole Lunny.

**Writing – original draft:** Carole Lunny, Lorri Puil, Tracy Liu, Savannah Gerrish, Douglas M. Salzwedel, Barbara Mintzes, James M. Wright.

**Writing – review & editing:** Carole Lunny, Lorri Puil, Tracy Liu, Savannah Gerrish, Douglas M. Salzwedel, Barbara Mintzes, James M. Wright.

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
