## [Decision Letter · Decision Letter 0]

26 Mar 2021

PONE-D-20-33171

Over half of clinical practice guidelines use non-systematic methods to inform recommendations: a methods study

PLOS ONE

Dear Dr. Lunny,

Thank you for submitting your manuscript to PLOS ONE. After careful consideration, we feel that it has merit but does not fully meet PLOS ONE’s publication criteria as it currently stands. Therefore, we invite you to submit a revised version of the manuscript that addresses the points raised during the review process.

Please adress the comments raised by reviewer 1. Furthermore, in the light that the search strategy is more than two years old, we think that it is important to discuss the current relvance and how the field may have evolved since 2018. In addtion, it migth be worth to discuss the future impact of the Rigth-Statement.

We look forward to receiving your revised manuscript.

Kind regards,

Tim Mathes

Academic Editor

PLOS ONE

Journal Requirements:

3. We noticed you have some minor occurrence of overlapping text with the following previous publication, which needs to be addressed:

https://pubmed.ncbi.nlm.nih.gov/31964662/

The text that needs to be addressed involves the Strengths and Limitations section and the caption of Figure 1.

In your revision ensure you cite all your sources (including your own works), and quote or rephrase any duplicated text outside the methods section. Given the extent of the overlap in the methods section, we ask that you make sure any directly reproduced text is quoted and cited clearly.

Further consideration is dependent on these concerns being addressed.

Reviewers' comments:

Reviewer's Responses to Questions

**Comments to the Author**

1. Is the manuscript technically sound, and do the data support the conclusions?

Reviewer #1: Yes

Reviewer #2: Yes

2. Has the statistical analysis been performed appropriately and rigorously? 

Reviewer #1: Yes

Reviewer #2: Yes

3. Have the authors made all data underlying the findings in their manuscript fully available?

Reviewer #1: Yes

Reviewer #2: Yes

4. Is the manuscript presented in an intelligible fashion and written in standard English?

Reviewer #1: Yes

Reviewer #2: Yes

5. Review Comments to the Author

Reviewer #1: Peer review comments

Thank you for the opportunity to review this

Interesting manuscript on guideline development. The authors have found that there are less guidelines using systematic methods in their development than their should be and other critical issues in guideline development. I thank the authors for conducting this work and have made some suggestions below.

Major

Minor

I think the statement regarding optimal methods for guideline development are lacking is somewhat misleading. Although different development organisations may have slight differences, there is consensus surrounding key aspects of guideline development (particularly in relation to the use of systematic reviews to inform their development) as evidenced by the IOM definition, the GIN standards, the GRADE approach, the GIN McMaster checklist, NHMRC guidelines for guidelines, NICE, AGREE II etc, etc. there are also reporting standards (RIGHT) for guidelines. I think the findings from this study are more surprising that DESPITE these standards and guidance, there are still many who do not use evidence synthesis.

A bit more information on selecting the sample size would be useful.

Reviewer #2: The publication by Lunny et al deals with a relevant topic. Although quality criteria and methodological standards for guidelines have been established for several decades, guidelines still exist that do not fully comply with these requirements and whose recommendations are not always consistently based on the necessary evidence or do not always adequately evaluate and interpret it.

The present study is based on transparent and comprehensible methods in terms of research, inclusion criteria, data extraction and analysis. It should be critically noted that due to the selection of sources for the search, the proportion of guidelines from the USA and Canada is very high in relation to guidelines from other countries. Concomitantly, non-English language guidelines are not analyzed, although many countries (including Germany) have well-developed guideline programs that result in evidence-based guidelines. However, in view of the effort involved, it is understandable that this route and this limitation were chosen. Also, a broader set of questions, including diagnosis and screening, would have been interesting, as there may be even greater deficiencies in evidence-based recommendations for these than for therapeutic questions.

Furthermore, the question arises why Cochrane Reviews per se seem to have a higher value than Non-Cochrane Reviews. There is no doubt that the methodological standard of Cochrane Reviews is generally very high, but there are also very good non-Cochrane Reviews (and also rather not so good Cochrane Reviews), so that it is quite understandable if these serve as a basis for a guideline recommendation, although a Cochrane Review exists. In addition to timeliness, contextual things may play a role here that are unlikely to be apparent without deeper analysis. In this context, a more in-depth discussion of the quality of the reviews would also have been desirable, but the authors themselves address this problem as a limitation.

Overall, this is a well-designed and well conducted analysis on a relevant topic. The conclusions can be derived from the analyzed data and limitations are addressed by the authors in the discussion.

6. PLOS authors have the option to publish the peer review history of their article (what does this mean?). If published, this will include your full peer review and any attached files.

Reviewer #1: **Yes: **Zachary Munn

Reviewer #2: **Yes: **Dr. Michaela Eikermann

---

## [Author Response · Author response to Decision Letter 0]

30 Mar 2021

We have attached our 'response to reviewers' as a Word document.

---

## [Editor Report · Decision Letter 1]

6 Apr 2021

Over half of clinical practice guidelines use non-systematic methods to inform recommendations: a methods study

PONE-D-20-33171R1

Dear Dr. Lunny,

We’re pleased to inform you that your manuscript has been judged scientifically suitable for publication and will be formally accepted for publication once it meets all outstanding technical requirements.

Kind regards,

Tim Mathes

Academic Editor

PLOS ONE
---

## [Editor Report · Acceptance letter]

8 Apr 2021

PONE-D-20-33171R1 

Over half of clinical practice guidelines use non-systematic methods to inform recommendations: a methods study 

Dear Dr. Lunny:

I'm pleased to inform you that your manuscript has been deemed suitable for publication in PLOS ONE. Congratulations! Your manuscript is now with our production department. 

Kind regards, 

on behalf of

Dr. Tim Mathes 

Academic Editor

PLOS ONE